# Testing for saturation in qualitative evidence syntheses: An update of HIV adherence in Africa

**Anke Rohwer**[1], **Lynn Hendricks**[1,2], **Sandy Oliver**[3,4], **Paul Garner**[5]*

**1** Centre for Evidence-Based Health Care, Division Epidemiology and Biostatistics, Department of Global Health, Faculty of Medicine and Health Sciences, Stellenbosch University, Cape Town, South Africa, **2** Social, Methodological, Innovative, Kreative, Centre for Sociological Research, Faculty of Social Sciences, Katholieke Universiteit Leuven, Leuven, Belgium, **3** EPPI-Centre, Social Research Institute, University College London, London, United Kingdom, **4** Africa Centre for Evidence, Faculty of Humanities, University of Johannesburg, Johannesburg, South Africa, **5** Centre for Evidence Synthesis in Global Health, Department of Clinical Sciences, Liverpool School of Tropical Medicine, Liverpool, United Kingdom

* Paul.Garner@lstmed.ac.uk

**Data Availability Statement:** All relevant data are within the manuscript and its Supporting information files.

**Funding:** This project was supported by the Research, Evidence and Development Initiative

## Abstract

### Background

A systematic review of randomised trials may be conclusive signalling no further research is needed; or identify gaps requiring further research that may then be included in review updates. In qualitative evidence synthesis (QES), the rationale, triggers, and methods for updating are less clear cut. We updated a QES on adherence to anti-retroviral treatment to examine if thematic saturation renders additional research redundant.

### Methods

We adopted the original review search strategy and eligibility criteria to identify studies in the subsequent three years. We assessed studies for conceptual detail, categorised as 'rich' or 'sparse', coding the rich studies. We sought new codes, and appraised whether findings confirmed, extended, enriched, or refuted existing themes. Finally, we examined if the analysis impacted on the original conceptual model.

### Results

After screening 3895 articles, 301 studies met the inclusion criteria. Rich findings from Africa were available in 82 studies; 146 studies were sparse, contained no additional information on specific populations, and did not contribute to the analysis. New studies enriched our understanding on the relationship between external and internal factors influencing adherence, confirming, extending and enriching the existing themes. Despite careful evaluation of the new literature, we did not identify any new themes, and found no studies that refuted our theory.

### Conclusions

Updating an existing QES using the original question confirmed and sometimes enriched evidence within themes but made little or no substantive difference to the theory and overall

(READ-It) project (project number 300342-104). READ-It is funded by aid from the UK government; however, the views expressed do not necessarily reflect the UK government's official policies. The funders had no role in study design, data collection and analysis, decision to publish, or preparation of the manuscript.

**Competing interests:** The authors have declared that no competing interests exist.

findings of the original review. We propose this illustrates thematic saturation. We propose a thoughtful approach before embarking on a QES update, and our work underlines the importance of QES priority areas where further primary research may help, and areas where further studies may be redundant.

## Introduction

Quantitative meta-analyses assume knowledge is cumulative, and thus a systematic review of randomised controlled trials may conclude that the question is answered, and no further primary research is justified [1]. Our experience with qualitative synthesis made us wonder whether, in qualitative research, we could reach a stage where we have enough evidence to answer a question and thus further primary research would be redundant. We had completed a qualitative evidence synthesis (QES) on adherence to anti-retroviral treatment (ART) in people living with Human Immunodeficiency Virus (HIV) in Africa [2]. At the time, we were surprised by the sheer volume of qualitative research that had been conducted: over 260 included studies, 76 identified as 'thick', defined in the original review as the "depth of analysis reflected in primary study authors' interpretation of findings." [2] Since this first edition, we have discussed and reflected on the meaning of 'thick' and 'thin' papers in qualitative research and found the description in the new chapter on qualitative evidence in the Cochrane Handbook useful [3]. According to this guidance, the terms 'thick' and 'thin' are "best used to refer to higher or lower levels of contextual detail", whereas the terms 'rich' and 'poor' are more appropriate to describe "qualitative evidence that includes higher or lower levels of conceptual detail". When examining saturation, conceptual richness of papers is more appropriate than contextual detail. Since the term 'poor' can also be associated with the methodological quality and is thus ambiguous, we used the term 'sparse' to describe studies with a low level of conceptual detail. We have thus used the terms 'rich' and 'sparse' to describe the depth of analysis in relation to conceptual detail (rather than contextual detail) in papers included in this study.

Our search date for the review was 2016, so we decided to update this review, and explore, in the rapidly moving policy area of HIV treatment and adherence in Africa, whether additional qualitative primary research is justified.

### Data saturation

Data saturation in primary research is defined as when "no new information or themes are observed in the data" [4]. It refers to the criterion which is used to stop data collection or analysis in primary qualitative studies. Saunders and colleagues (2018) describe four models of saturation in qualitative research [5]:

1. theoretical saturation, which "relates to the development of theoretical categories" and judging when to stop data collection based on whether data collected has covered diverse categories;

2. inductive thematic saturation, relates to "the emergence of new codes or themes", and additional data does not lead to any new themes (focus on data analysis);

3. *a priori* thematic saturation, relates to "the degree to which identified codes or themes are exemplified in the data", a more deductive approach, where a theory is prespecified and the data is collected to illustrate the theory (related to sampling);

4. data saturation, related to "redundancy in the data", when new data just repeats what has already been found in previous data (related to data collection).

Data saturation in qualitative evidence synthesis is a relatively new concept. France and colleagues (2016) argue that in cases where the original QES had reached 'conceptual saturation', it is unlikely that new studies will provide new insights. In quantitative synthesis, when there is a substantive body of knowledge around a topic and no change of circumstances, researchers generally agree that further research on the topic is redundant and represents a waste of resources. We note that researchers are applying "updating" principles of quantitative research synthesis to QES [6–8]. However, we were concerned about wholesale transfer of these principles to QES, and wanted to think through the methods, consider sensible criteria to consider triggers for updates, and whether indeed the concepts of saturation in primary research apply to synthesis.

## Materials and methods

Our objective was to update the review with the additional three years of data; and then step back and use formal appraisal to examine the changes that the update had on the findings of the original review. Our approach was to evaluate data saturation in relation to coverage against the original conceptual model; and whether new studies confirmed, extended, enriched or refuted the themes in the final review.

We identified new studies that met the same eligibility criteria as in the first version of this review [2]. We considered qualitative studies conducted in low-and middle-income countries (LMICs) that explored perspectives, perceptions and experiences of people living with HIV (PLHIV), caregivers and healthcare providers on linkage to and retention in HIV care, as well as adherence to ART. We used the same search strategy as per the first version of this review [2] for studies published from December 2016 (the date of the last search of the original review) to 8 November 2019. All authors were authors on the original review. Two authors (AR and LH) took the same hands-on role they held in the original review and were familiar with the meaning and operational aspects of inclusion criteria and existing codes. They independently screened titles and abstracts of the search output using Covidence software [9] and retrieved and independently screened full texts of potentially relevant studies.

From eligible studies, we purposively selected all studies conducted in Africa as per the original review, and two authors (AR and LH) independently assessed them as being 'rich' or 'sparse', referring to higher or lower levels of conceptual detail within the study [3]. We considered the same features as per the original review when assessing the richness of papers: "1) the extent to which the authors transformed or analysed their findings (beyond lists of barriers and facilitators), 2) insight into participants perspectives was demonstrated, 3) richness and complexity had been portrayed (variation explained, meanings illuminated), and 4) theoretical or conceptual development" [2]. We intended to use these features as a checklist and provide a score for each of these features (yes, somewhat, no). However, when piloting the checklist, we did not have good agreement when considering the scores, although we had good agreement on whether the paper should be categorised as 'rich' or 'sparse'. Moving forward we thus used the four features as a guide and the two authors independently assessed papers, discussed and decided on the category by consensus. Where consensus could not be reached, we discussed this further with the other authors. We included all rich studies in this updated review and extracted descriptive data for sparse studies. We would have sampled sparse studies if they covered any additional population groups not covered by the rich studies. The two authors (AR and LH) independently read and coded included studies. We coded studies using the existing

list of codes and adding to the list of codes if new codes emerged. For each new code we noted whether it:

1. confirmed the findings of our original review,

2. extended the existing themes,

3. deepened or enriched our understanding of existing themes by providing illustrative examples or rich descriptions of experiences, or

4. refuted the findings of our original review.

When we developed these criteria, we also intended to examine if the studies crystallised our understanding of the themes. 'Crystallised' we thought at the outset would refer to understanding within themes, but as we advanced we realised crystallised was a good term, but actually the data helped crystallise at a higher level, in terms of understanding our theory, rather than connections within themes.

The two authors discussed and agreed on coding and findings for each study. The whole team met regularly to discuss new codes, and to identify any new themes or sub-themes. Through this iterative process we also discussed the impact of new codes on our proposed theory and conceptual model.

We report the number of studies confirming our original findings, extending existing themes, enriching our understanding of existing themes, and refuting our theory; and summarise the updated findings in relation to the findings from the original review. We then discuss our experience with updating the review as it relates to data saturation.

## Reflexivity

We are all interested in qualitative research synthesis and excited by its potential. We have all contributed to Cochrane, an organization explicit about a cumulative model of evidence synthesis. We bring clinical and academic backgrounds including nursing and midwifery (AR), infectious diseases (PG) and social sciences (LH and SO) and recognize that these will influence how we interpret the evidence. Furthermore, all authors contributed to the original review [2] and as such may be less open to the emergence of new themes or altering our conceptual model. On the other hand, we all believed at the outset that new findings might arise from: changes with time, with society, with policy, or changes in public opinion, or other external unpredictable factors that we could not anticipate. It was this potential for new knowledge that was an impetus to us to evaluate saturation and update the review. This led to a much more open discussion about whether new themes or alteration were needed to encompass the new research. All authors view data saturation in primary research as a point in data collection where additional data does not add any new insights [5] and believe that this is achievable. We were unsure whether the same principle applies to QES, which is why we undertook this study.

## Results

### Results of the search

We identified 301 studies meeting the inclusion criteria, after screening 3895 titles and abstracts and examining 352 full texts. Seventy-three studies were not conducted in Africa and thus not included in the sample. We assessed 82 papers as being rich and included them in the analysis (Fig 1). We did not sample any sparse papers.

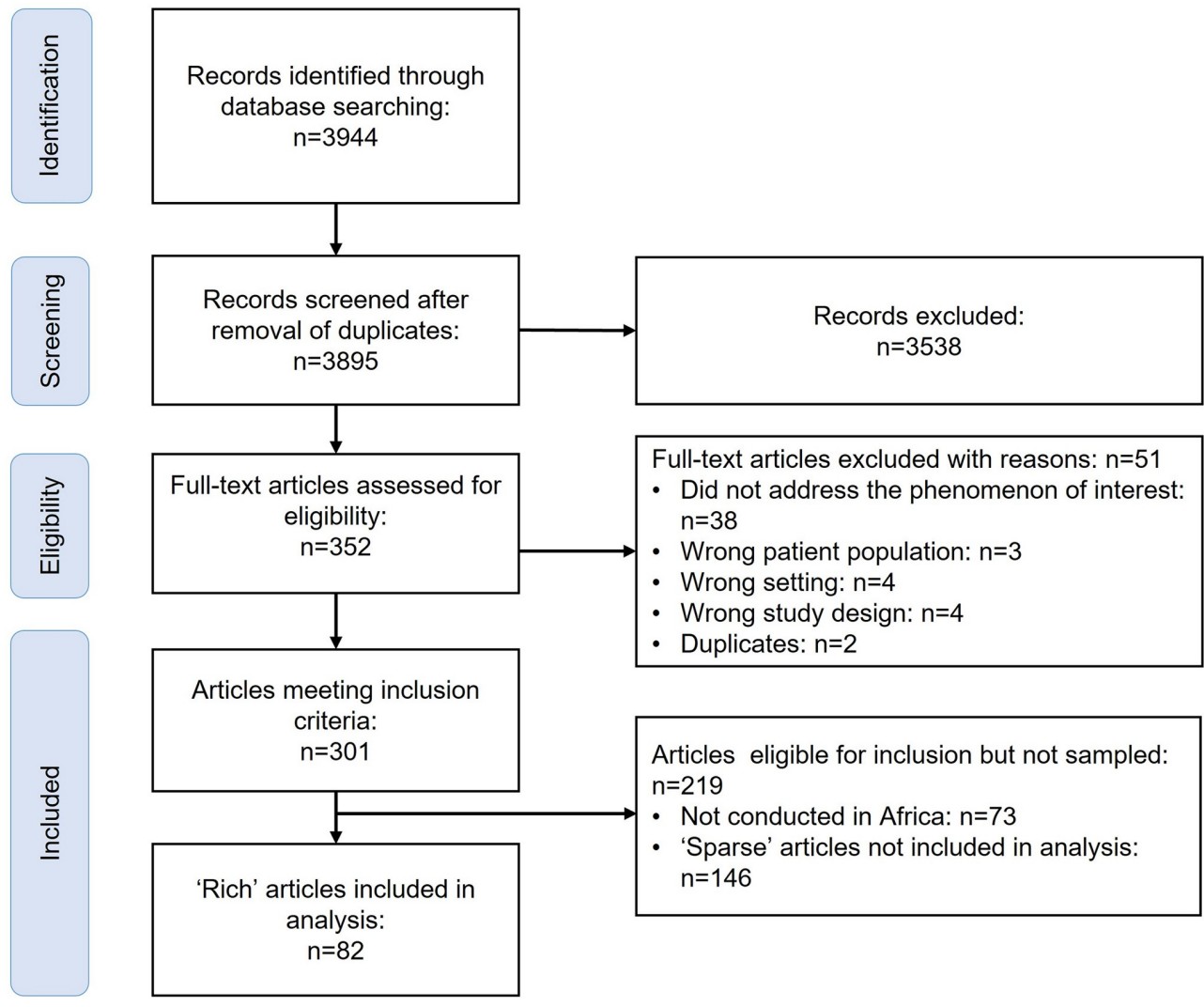

**Fig 1. PRSIMA flow diagram.**

### Description of included 'rich' studies

The 82 rich studies [10–91] were conducted in South Africa (n = 20), Uganda (n = 12), Kenya (n = 9), Zambia (n = 6), Malawi (n = 4), Tanzania (n = 3), Eswatini (n = 3), Côte d'Ivoire (n = 3), Zimbabwe (n = 1), Mali (n = 1), Democratic Republic of Congo (n = 1), Nigeria (n = 1), Ethiopia (n = 1), Rwanda (n = 1), and Ghana (n = 1). One study was conducted in Kenya, Malawi and Mozambique, one in Malawi and Zimbabwe, and one in Ghana, Uganda and Zambia. One study was conducted in the UK, Ireland, USA and Uganda, but we only considered data relevant to Uganda for this update. The 'Bottlenecks study' is a multi-country study [92] conducted in Malawi, Uganda, Tanzania, Kenya, Zimbabwe and South Africa and is reported in nine included papers.

Studies evaluated factors influencing linkage, retention in care and adherence to ART among adult PLHIV (n = 32), adolescents or children living with HIV (ALHIV) (n = 14), men (n = 8), women (n = 2), pregnant or postpartum women (n = 9), sero-concordant or sero-discordant couples (n = 7), men who have sex with men (MSM) (n = 3), alcohol and other drug

**Table 1. Number of studies confirming, extending and enriching the findings of the original review.**

| Themes in 2019 analysis | Number of studies | | |
|---|---|---|---|
| | Confirming | Extending | Enriching |
| 1. Poverty, competing priorities and an unpredictable microworld | 27 | 0 | 12 |
| 2. Social identity and gender norms can have a profound impact on care-seeking behaviour | 33 | 9 | 24 |
| 3. Alienation makes it hard to take ART | 24 | 0 | 10 |
| 4. People with HIV receive conflicting information, messages and views | 20 | 5 | 13 |
| 5. "Bad patients" are an unhelpful construct of an authoritarian health | 21 | 1 | 12 |
| 6. Poor clinic services for patients and inadequate support for health workers | 13 | 3 | 8 |
| 7. The new normal requires daily drugs | 27 | 7 | 17 |
| 8. Self-efficacy, social responsibility and support helps | 42 | 9 | 26 |
| 9. The tipping point | 6 | 0 | 9 |

users (n = 2), older PLHIV (n = 2), people living with disabilities and HIV (n = 1) and migrants (n = 1). Sparse papers (n = 146) did not include any additional subgroup of participants not covered in the rich papers and therefore did not contribute to the analysis. Rich studies used a variety of data collection methods, with semi-structured, in-depth interviews and focus group discussions being the most common methods. Thirty-nine studies used more than one method to collect data. Study participants were mostly HIV positive, but also included caregivers of HIV positive children or adolescents, healthcare providers, traditional healers and some HIV negative participants. S1 Table contains a summary of the characteristics of all 'rich' papers.

Rich papers included in this updated version of the review confirmed our initial findings for all themes, extended our findings for themes 2, 4, 5, 6, 7 and 8, and deepened and enriched our understanding of all themes (Table 1). New studies particularly enriched findings related to themes 2, 4, 7 and 8. None of the included studies refuted our theory. References to the studies confirming, extending and enriching the themes are provided in S2 Table.

## Updated review findings

Below we summarise the original findings, and how the new studies relate to this. New codes and subthemes are listed in S3 Table.

**Theme 1: Poverty, competing priorities and an unpredictable microworld.** *Original findings*. Many PLHIV live in poverty. Needs related to basic survival such as earning money to buy food and safety are often prioritized over attending health services. For some, maintaining a social life is more important than engaging in care and adhering to treatment. In addition, many PLHIV experience unpredictable events that disrupt their lives and lead to disengagement.

*Updated findings*. New studies offered further examples that ***confirmed*** the significance of competing priorities [38, 39, 41, 61, 67, 71, 72, 80, 89] and unpredictable life events [15, 53, 66]. Priorities include going to school and writing exams, seeking care for other chronic diseases such as diabetes of hypertension, and meeting religious obligations. Unpredictable life events include caring for premature, sick or hospitalized children, treatment failure and sudden illness.

**Theme 2: Social identity and gender norms can have a profound impact on care-seeking behaviour.** *Original findings*. Gender roles interact with HIV care. Many men experience HIV as a threat to their masculinity and find it difficult to both fulfil hegemonic features of masculinity and engage in HIV care. Women have limited choices when making healthcare decisions, as they have defined social roles, are dependent on their partners and may

experience sexual abuse and intimate partner violence (IPV). Adolescents living with HIV experience a lot of confusion and isolation, want to fit-in with their peers and need a lot of support to navigate their illness and future aspirations. HIV positive people with disabilities experience multiple stigmas and discrimination.

*Updated findings*. We ***extended*** this theme to include relationship dynamics between sero-concordant and sero-discordant couples that influence engagement in care and adherence to ART [16, 17, 19, 36, 37, 69, 83]. Most couples desire long-term relationships and marriage despite some finding it difficult to disclose their status and others blaming each other for bringing HIV into their lives. Power dynamics within relationships often change according to HIV status. Women generally lose power if they are diagnosed with HIV, but gain power if men brought HIV into the relationship or died. In sero-concordant couples, power is often shared between both partners, resulting in a more harmonious life and better engagement in care.

Our understanding of adolescents' experiences with HIV and ART was ***enriched*** considerably [11, 12, 26, 38, 41, 54, 56, 63, 69, 87]. Generally, adolescents living with HIV do not receive the support they need to understand their illness and the impact of HIV on their future, engage in care and adhere to treatment. They need to navigate a life filled with secrets and silence. Many feel left out, disempowered, angry, depressed and have suicidal ideations. Most adolescents experience multiple losses, including the loss of their parents, caregivers and their social environment (schools, neighbourhoods, friends, relatives) which has a profound impact on their mental health. They often do not have the opportunity to grieve these losses, and battle to come to terms with them. This is partly due to the lack of disclosure of their own and their parents' status and lack of knowledge about HIV, which is still seen as a death sentence for many. This leads to fear and confusion, as their concerns are generally not discussed at home or at the clinic. Indeed, this 'culture of silence' was a prominent theme across studies. In addition, some adolescents experience violence, sexual assault, abuse and neglect by their parents.

We ***extended*** this theme to include older people living with HIV [13, 39, 67]. Many older people feel insecure, outnumbered and considered already dead. They need to deal with stigma from the general population, but also from healthcare providers and feel that they are being treated differently due to their age. They are a unique group of people, as they have experienced HIV as a disease that causes death, lived through the numerous advancements in care and have witnessed how ART has changed the narrative of HIV being a deadly disease, to a chronic condition. This makes it easier for them to accept their diagnosis and motivates them to remain in care and adhere to treatment.

**Theme 3: Alienation makes it hard to take ART.**   *Original findings*. HIV stigma and discrimination undermine a sense of belonging, which impairs adherence to treatment and engagement in care. PLHIV fear inadvertent disclosure and may therefore disengage from care or skip doses if confidentiality is at risk.

*Updated findings*. New studies ***enriched*** our understanding of how various forms of stigma affect the daily lives of PLHIV [11, 13, 14, 34, 38, 46, 68, 69, 76, 77]. Anticipated stigma and the fear of having a positive HIV status revealed, leads to feelings of shame, humiliation and suicidal ideations. Studies reported that perceived stigma, or the fear of being stigmatised is the most prevalent form of stigma experienced by adolescents. They fear social isolation, losing friends, diminished social interactions and loss of respect, but also loss of material support such as housing, food and employment. Although stigma affects all PLHIV, some subgroups (adolescents, older people, people living with disabilities and migrants) experience more severe forms of discrimination than others. When faced with any form of stigma, most PLHIV go through a wide range of complicated emotional responses such as shame, fear, confusion, guilt, anger and grief, that undermine a sense of belonging. HIV is a biosocial disease, where

clinical knowledge and social perceptions of the disease are constantly changing. Among many PLHIV, this has led to a shift from fear of physical death to fear of social death.

**Theme 4: People with HIV receive conflicting information, messages and views.** *Original findings.* Alternative discourses, beliefs, and various sources of information about HIV and its treatment caused uncertainty about ART and engaging in care. PLHIV have to navigate through the information and their experiences to choose an ideology that suits their needs best. Consulting traditional healers was often preferred to the biomedical approach.

*Updated findings.* New studies ***enriched*** our understanding of cognitive dissonance and individual beliefs that influence engagement in care [10, 14, 18, 32, 38, 41, 42, 46, 51, 59, 64, 68, 77, 80, 81, 86, 91]. Although knowledge about HIV generally increases over time, social perceptions and individual beliefs seem to have a greater influence on how PLHIV engage in care. PLHIV are faced with concurrent and co-existing therapeutic alternatives (medical pluralism) and need to choose from this wide range of options that includes biomedical, traditional and faith-based approaches. Their choice depends on a number of factors including cultural and familial traditions, individual perceptions of illness, past experiences and religious views. In this update, findings suggested that some PLHIV favoured the biomedical approach due to the rigorous medical processes, low cost and perceived effectiveness of the treatment, while believing that traditional healers were dishonest, ineffective and only cared about the profit they made.

**Theme 5: "Bad patients" are an unhelpful construct of an authoritarian health system.** *Original findings.* HIV care is delivered within an authoritarian health system, where health care workers hold all the power and expect PLHIV to abide by the rules and guidelines of positive living. Some PLHIV manage to adhere to the rules, while those that are unable to are judged, blamed, punished and labelled 'defaulters' or 'bad patients'. This leads to feelings of shame, guilt and stress.

*Updated findings.* New studies ***enriched*** the analysis with additional detail of how PLHIV and HCWs perceive each other [12, 32, 35, 40, 45, 49, 51, 52, 55, 60, 62, 88]. Many PLHIV want to be part of the treatment decision-making process. However, they feel that their voices are not being heard as HCWs see themselves as the 'knowledgeable healers' and PLHIV as the 'ignorant sick'. PLHIV feel that instead of reprimanding and punishing PLHIV, HCWs should build relationships and provide advice and health education that is non-discriminatory, non-judgemental, extends beyond the rules of 'positive living' and focuses on patients' concerns and desire to lead a normal life.

We ***extended*** this theme to reflect how competing gender and professional norms can hinder HIV care [40]. Many male HCWs are accustomed to positions of clear authority and social power, and this expectation sometimes colours how they offer care to female patients. Some male patients, on the other hand, sometimes refuse to recognise and respect clinical knowledge and authority from female HCWs. Furthermore, female HCWs are sometimes exposed to inappropriate and unwanted sexual attention from male patients. Gender can therefore inhibit HCW's ability to establish professional boundaries.

**Theme 6: Poor clinic services for patients and inadequate support for health workers.** *Original findings.* Attending HIV services is stressful and unpleasant due to disrespectful HCWs, inflexible services, long waiting times and rigid policies. HCWs feel over-burdened amidst staff shortages, lack of resources, inadequate training and support, and pressure to produce good outcomes.

*Updated findings.* New studies added further examples that ***confirmed*** negative experiences at HIV clinics [31, 32, 45, 48, 49, 52, 55, 57, 60, 79]. Physical space and layout of HIV clinics is not optimal due to poor signage and poorly laid out waiting areas (inside and outside the clinic) which do not offer PLHIV the privacy they desire. HIV services do not accommodate

needs of vulnerable groups such as adolescents. Referral and collaboration between clinic and community services were also reported as being poor.

**Theme 7: The new normal requires daily drugs.**    *Original findings*. It is difficult to accept HIV and ART. While some feel depressed and struggle to cope, others manage to accept the diagnosis and incorporate ART into their daily lives. Health status at time of diagnosis plays an important role in motivating people to adhere to treatment and engage in care.

*Updated findings*. New studies provided further examples ***confirming*** how PLHIV adapt to the 'new normal' and ***enriching*** our understanding of how they deal with this [12, 14, 15, 19, 29, 33, 34, 42, 47, 49, 62, 63, 65, 69, 70, 75]. PLHIV who have not accepted their diagnosis, generally find it difficult to adhere to ART and engage in care. Some use avoidance tactics such as discarding ART, switching clinics, or providing false information. Others give up hope completely and reach a level of apathy, where they don't care about treatment and wait to die. Improvement in health status can lead to sustained adherence in some PLHIV, while in others, it leads to decreased motivation to continue ART. PLHIV who have experienced first-line treatment failure, may see second- and third-line treatment as a second chance and a catalyst for incorporating ART into their lives.

We ***extended*** this theme to include the enabling effects of ART [13, 33, 34, 63, 65, 76]. PLHIV described various positive effects of ART such as being able to live a meaningful life, caring and providing for one's family and being independent. For men, ART can restore a sense of masculinity that includes having physical strength to work, being able to fulfil their role as head of the household and having sexual relationships. ART can also reduce stigma as it prevents unintended disclosure once people experience improvement in health status. Some people even manage to resist stigma and disclose their status openly, giving them the freedom to live a normal life. ART has psychological benefits that include reducing stress, improving self-image and restoring a sense of belonging. Over the years, ART has also led to increased societal acceptance of HIV in communities.

**Theme 8: Self-efficacy, social responsibility and support helps.**    *Original findings*. People vary how they respond to HIV and life-long ART, some have high self-motivation, while others have very little. Emotional, financial and practical support from other people, including healthcare workers and family, helps PLHIV cope with their diagnosis and encourages continued engagement in care.

*Updated findings*. We ***extended*** this theme to include resilience and its importance in overcoming difficulties and adapting to lifelong ART [11, 25, 29, 30, 57, 63, 69, 73, 81]. Social capital in form of social networks, is an important resource to promote resilience as it can facilitate access to resources such as food, transport, and ART; and encourages adherence and engagement in care. Resilience promotes self-acceptance and acknowledgement of self-worth. It also enables PLHIV to plan a future, help others and persevere when faced with setbacks. Feeling hopeful helps to remain in care.

New studies also ***confirmed*** this theme with more detailed examples of various forms of support that can help PLHIV cope with lifelong ART [14, 16–20, 26, 35, 36, 43, 45, 47, 49, 56, 57, 61, 66–68, 76, 81, 82, 84, 89–91]. Health services can be more supportive by offering adolescent-friendly clinics, mobile services, increased ART days, free services, and improved access to care. HCWs who are supportive and respectful create a strengthened sense of connectedness with PLHIV. This includes partnering with patients to personalise medical regimes, talking through conflicts with family members, linking with counsellors or support groups and facilitating transfers to other facilities if needed. Support from healthcare workers can also include practical support for transport and food, as well as using connections to find drugs during drug stock-outs. Most PLHIV value healthcare workers that go beyond what is expected of them, those that are friendly and welcoming, competent, able to reassure patients, explain HIV

in simple terms, maintain confidentiality and act in a non-discriminatory way. This helps to build a trusting relationship between PLHIV and healthcare workers. Support from peers is particularly important for defined populations, including adolescents, people who use drugs or alcohol, or MSM. Some PLHIV actively participate in peer support groups, while others are passive participants. People living openly with HIV act as role models and have an important role in educating, communicating and empowering other PLHIV. Partners can provide instrumental, informational and emotional support. Some children provide emotional support to their parents, help them to self-manage their condition and provide material resources such as food and shelter.

**Theme 9: The tipping point.** *Original findings*. There are several internal and external influences that interact and lead to disengagement or reengagement in care. Some PLHIV experience a final event or 'tipping point' that leads them to disengage or reengage in care.

*Updated findings*. New studies *enriched* our understanding of the complex and dynamic interplay between various influencing factors and time, that leads to engagement or disengagement in care [15, 16, 34, 35, 47, 61, 62, 73, 81]. These constantly changing factors include knowledge and understanding of HIV and ART, the cyclical relationship between adherence and relationship dynamics, resources, treatment failure, health status, social hierarchies, family responsibilities and relationships with HCWs. PLHIV weigh up the benefits and risks of treatment-taking and status exposure on a daily basis, and thus constantly make conscious or unconscious decisions about adherence. HIV diagnosis, treatment and moving from illness to health; as well as coping and adapting to the new normal are not linear processes. Rather, they represent a continuum from illness to taking and adhering to ART and feeling healthy; and from coping poorly to reaching a sense of belonging and adapting to the new normal. PLHIV constantly move up and down this continuum while cycling in and out of care.

## Discussion

We updated our QES on factors influencing ART adherence. We included 82 rich papers, and this analysis confirmed, extended, and enriched our understanding of existing themes. None of the studies refuted our theory, despite careful consideration within the team of possible adjustments being made. The updating was a major undertaking.

### Summary of impact of the update on the review findings

The findings *confirmed* the theoretical understanding of factors influencing adherence to ART. This included additional examples confirming: the significance of competing priorities and unpredictable life events; negative experiences at HIV clinics; and the value of support for adherence and the role health services play.

The findings *extended* understanding of power: within relationships for sero-concordant and sero-discordant couples; for older people living with HIV; and between practitioners and patients adopting gendered norms. The findings also *extended* understanding of the enabling effects of ART for restoring role norms, for reducing stigma and stress, and for increasing societal acceptance of HIV.

The findings *enriched* understanding in several themes: of adolescents' lives and the damaging effects of a 'culture of silence'; of how various forms of stigma affect the daily lives of PLHIV; of cognitive dissonance and individual beliefs relating to biomedicine and traditional healing; of tensions within the treatment decision-making process; and of adapting to the new normal of living with HIV. In addition, the new studies *enriched* our understanding of the complex and dynamic interplay between various influencing factors and over time.

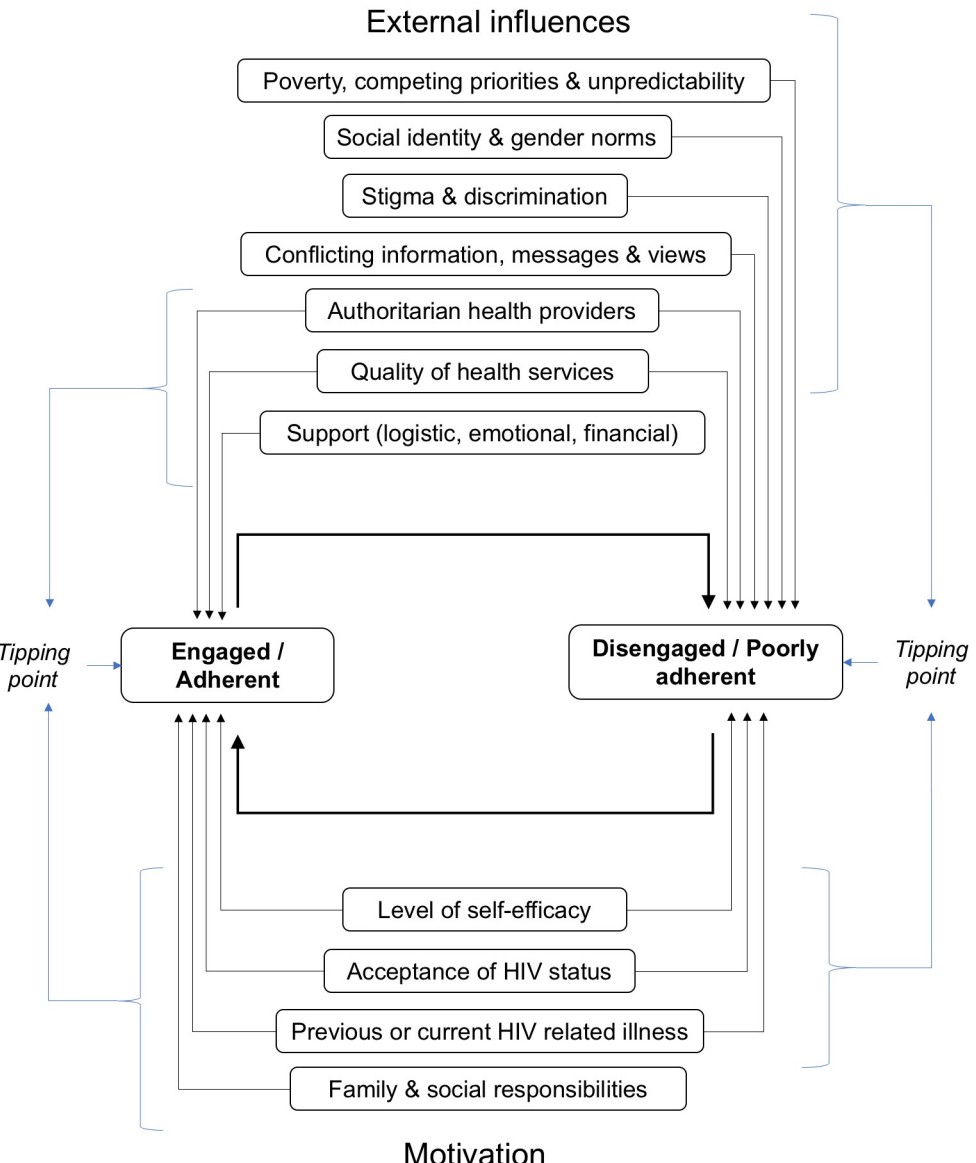

**Fig 2. Conceptual model of multiple influences on engagement and adherence behaviour [2].**

No new themes emerged from the updated analysis. Within the author team, we discussed changing the name of Theme 7 and splitting Theme 8 into two themes, but after careful consideration, we concluded that it made more sense to enrich and extend the original themes. The extra details, illustrative examples and in-depth descriptions of experiences contained in the new studies allowed us to paint a clearer, more colourful, and more comprehensive picture of how PHLIV understand and respond to pressures they encounter in their daily lives. We were thus able to gain greater insight into factors that influence the dynamic process of engagement, disengagement and reengagement in care, and how these factors are connected (Fig 2). In hindsight, we recognise Theme 9, 'The tipping point', as a crystallisation of everything described in Themes 1–8.

## Observations of the methodological implications

We were surprised at the number of additional studies that met our inclusion criteria (n = 301) published in a three-year period and wondered whether these would reflect changes over time, such as HIV policy changes, changes in public opinion or society, or other unpredicted factors that might influence adherence; or indeed, if the existing research would drive new studies and lead to new themes. However, included studies did not report on any substantial changes. We also expected that new studies would cover additional population groups or contexts, especially if marginalized. We identified studies of new population groups, such as sero-concordant and sero-discordant couples, in our analysis of rich papers, although these could appear in the main text but not the abstract. Whilst in our methods we stated we would have used the sparse papers if they cover additional population groups, these groups were not studied in these papers.

Our updated analysis did not change the theory we proposed in the first version of the review but strengthened it considerably. Just as additional data in a meta-analysis would narrow the 95% confidence interval around the effect estimate to increase precision and confidence in the result, additional studies in our QES added more connections between data and themes and therefore increased the confidence in and validated our original theory. It raises the question whether updating a QES which already has a substantive literature base should be done without very considerable thought about whether this is a sensible use of resources. Rather, author teams should consider whether there is a new question that emerged from or builds on the existing review. This is in contrast to the guidance on updating QES provided by France and colleagues [6]. For QES with a less substantive literature base, we suggest author teams may carry out a scoping exercise of new studies to see whether an update would be valuable.

We were all authors on the previous review and thus had a good grounding in the theory, and our opinion is that, if updating is regarded as needed, then using the existing team is economical and sensible. Our understanding of the data and existing themes matured over time, allowing us to see the connections between themes even clearer. However, we acknowledge that adding new authors to the team could have enhanced our analysis by introducing different perspectives, particularly on differences in the data [93].

We found two other examples of updated QES. One review was conducted by a different author team [7], who adapted the question of the original review to include a wider range of eligible participants, from which we would anticipate extending the original findings. They analysed data of all papers included in the original and updated search. According to France and colleagues' guidance on updating meta-ethnography [6], this process is compared to knocking down and rebuilding an existing house. In the other example [8], the author team and the review question remained the same. However, authors were aware of additional studies conducted in similar patient groups, but in different contexts. In addition, they were aware of policy changes in Europe that might affect their findings. Authors included seven new studies in addition to the seven older ones, reclassified some of their themes and added a new theme. According to France and colleagues' analogy, this process can be compared to extending and renovating the original house.

Our original synthesis and our update included studies that presented findings rich in conceptual detail [3] with the aim of building a theory. We used the same approach to sample rich papers in both publications and considered four features that relate to rich findings, as described in the Methods section. Although these features did not work well as a formal checklist, we had good agreement when deciding whether a paper presented rich or sparse findings. Ames and colleagues [94] developed a 5-point scale to assess richness of data and for their QES

sampled papers that scored a 4 ("A good amount and depth of qualitative data that relate to the synthesis objective") or 5 ("A large amount and depth of qualitative data that relate in depth to the synthesis objective"). These descriptions fit well with the rich papers that we included in this update.

In our previous review we had sampled sparse studies to consider if we were "missing" themes by not including them and found that this was not the case. This provides solid justification for not including the 146 new sparse studies that did not contribute to the analysis. Although not contributing to this evidence synthesis, these sparse studies may fulfill other more immediate purposes such as informing specific programmes.

We applied the same principle of presenting rich findings to the original and updated synthesis, and only included findings if we had high confidence in them. We therefore had no opportunity to consider whether confidence in particular themes was increased, as might happen if studies were conceptually sparse rather than conceptually rich as we would not include them if there was doubt as to their validity. The new studies included in this update confirmed our confidence in our original theory. We did not use the GRADE-CERQual approach which is designed for assessing individual descriptive findings, as it is not currently known how well it performs when assessing higher order analytical findings and theory building [3, 95]. We are exploring ways to use the GRADE-CERQual approach on this dataset, and this is a separate piece of work in progress.

## Conclusion

Our updated QES of adherence to ART confirmed our confidence in the theory proposed in the original version of the review, and at times also extended and enriched existing themes. Despite our best efforts, we did not find any new themes in the rich papers, and this demonstrates a degree of data saturation, and research redundancy in a large number of sparse papers that did not contribute to the analysis. We do not see saturation as binary, but our findings mean updating should be thought through carefully before starting. Unless there is a very good reason to update a review asking the same question which is already well populated with rich primary studies, we believe it may be better to build on the knowledge generated by the existing review to develop synthesis around particular problematic or priority areas. It also underlines the role of high quality, timely QES to help identify and prioritise areas where further primary research may help, and areas where further studies may be redundant.

For this synthesis, which identified no new themes, thematic saturation (when no new codes or themes are emerging from the data) was achieved with the original synthesis even though some subsequent studies address new groups or contexts. This may be because all the included studies, in the original and subsequent syntheses, were conceptually rich. However, thematic saturation is not the same as saturation of meaning, when no additional information emerges from the quality, deep, detailed and relevant data that has been gathered [96]. Changes in meaning may arise in the literature from changes in policies or norms (for instance, legal changes relating to knowingly exposing others to HIV, or HIV/AIDS being more widely understood as a treatable disease). Alternatively, changes in meaning could arise from primary research teams or review teams imposing new interpretations. We therefore propose that more may be learnt from updating a QES when new studies have addressed new contexts (including contexts that have changed over time, for instance because of changes in policies or norms); adopted different interpretations or theories; or reported conceptually richer findings. In addition, more may be learnt if the review team that brings different experience and so may offer new sensitivities and insights during the synthesis.

## Supporting information

**S1 Table. Table of rich included studies.**
(DOCX)

**S2 Table. References to studies confirming, extending and enriching the findings of the original review.**
(DOCX)

**S3 Table. List of existing and updated themes, subthemes and codes.**
(DOCX)

**S1 Checklist. ENTREQ checklist.**
(DOCX)

## Acknowledgments

We would like to thank Anel Schoonees for conducting the updated search.

## Author Contributions

**Conceptualization:** Anke Rohwer, Lynn Hendricks, Sandy Oliver, Paul Garner.

**Data curation:** Anke Rohwer, Lynn Hendricks.

**Formal analysis:** Anke Rohwer, Lynn Hendricks, Sandy Oliver, Paul Garner.

**Funding acquisition:** Paul Garner.

**Investigation:** Anke Rohwer, Lynn Hendricks, Sandy Oliver, Paul Garner.

**Methodology:** Anke Rohwer, Lynn Hendricks, Sandy Oliver, Paul Garner.

**Project administration:** Anke Rohwer.

**Resources:** Lynn Hendricks, Sandy Oliver, Paul Garner.

**Supervision:** Sandy Oliver, Paul Garner.

**Validation:** Anke Rohwer, Lynn Hendricks, Sandy Oliver, Paul Garner.

**Visualization:** Anke Rohwer, Lynn Hendricks, Sandy Oliver, Paul Garner.

**Writing – original draft:** Anke Rohwer, Lynn Hendricks, Sandy Oliver, Paul Garner.

**Writing – review & editing:** Anke Rohwer, Lynn Hendricks, Sandy Oliver, Paul Garner.

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
