## [Decision Letter · Decision Letter 0]

9 Aug 2021

PONE-D-21-10801

Testing for saturation in qualitative evidence syntheses: An update of HIV adherence in Africa

PLOS ONE

Dear Dr. Garner,

Thank you for submitting your manuscript to PLOS ONE. After careful consideration, we feel that it has merit but does not fully meet PLOS ONE's publication criteria as it currently stands. Therefore, we invite you to submit a revised version of the manuscript that addresses the points raised during the review process.

Two accomplished scholars with expertise in qualitative evidence synthesis reviewed your manuscript. I also reviewed the paper, and I concur with the vast majority of the comments, critiques, and suggestions raised by the reviewers. While this is an important manuscript that makes a substantial contribution to the field of qualitative research synthesis, a few changes need to be made before being considered for publication.

If you would like to change your financial disclosure, please include your updated statement in your cover letter. Guidelines for resubmitting your figure files are available below the reviewer comments at the end of this letter.

If applicable, we recommend that you deposit your laboratory protocols in protocols.io to enhance the reproducibility of your results. Protocols.io assigns your protocol its identifier (DOI) to be cited independently in the future. For instructions, see: http://journals.plos.org/plosone/s/submission-guidelines#loc-laboratory-protocols. Additionally, PLOS ONE offers an option for publishing peer-reviewed Lab Protocol articles, which describe protocols hosted on protocols.io. Read more information on sharing protocols at https://plos.org/protocols?utm_medium=editorial-email&utm_source=authorletters&utm_campaign=protocols.

We look forward to receiving your revised manuscript.

Kind regards,

Sergi Fàbregues

Academic Editor

PLOS ONE

Journal Requirements:

Reviewers' comments:

Reviewer's Responses to Questions

**Comments to the Author**

1. Is the manuscript technically sound, and do the data support the conclusions?

Reviewer #1: Yes

Reviewer #2: Yes

2. Has the statistical analysis been performed appropriately and rigorously? 

Reviewer #1: N/A

Reviewer #2: N/A

3. Have the authors made all data underlying the findings in their manuscript fully available?

Reviewer #1: Yes

Reviewer #2: Yes

4. Is the manuscript presented in an intelligible fashion and written in standard English?

Reviewer #1: Yes

Reviewer #2: Yes

5. Review Comments to the Author

Reviewer #1: Thank you for a really interesting paper. Below are some comments which I feel could strengthen the content. This is an important contribution to a rapidly evolving QES field and addresses two important qeustions of saturation and when to update a review.

Abstract:

In the paper you talk about a certain type of richness (conceptual), can this be added to the abstract to clarify.

Introduction:

Can the authors provide more of a definition of what they mean by rich (thick) and poor. In the introduction thick is used as a synonym for rich but can have a different interpretation when linked to thick description. I suggest changing line 53 to rich.

There is a debate in the social science community about whether saturation is reachable in primary research. See, for example:

Braun 2019 (https://www.tandfonline.com/doi/full/10.1080/2159676X.2019.1704846?casa_token=yW3J1lQL49AAAAAA%3AtX3ULeoVVftcerWwaF2kmrsBv8d7F6B7Vt0q24uWoWZFszfaKYL10iqxOqCKVBJ9ojnj6CMzF9UP2e0)

Low 2019 (https://www.tandfonline.com/doi/full/10.1080/00380237.2018.1544514?src=recsys)

Sebele-Mpofu 2020 (https://www.tandfonline.com/doi/full/10.1080/23311886.2020.1838706)

I think this same debate can apply to secondary data analysis like in a QES. I suggest the authors mention this debate and state in the reflexivity section that they believe data saturation is achievable (or not).

Line 76- The authors write that QES are adopting methods of quantitative research synthesis- I don’t think this is completely correct. We are adopting some similar approaches in relation to the steps in a SR but the methods for a QES are quite different to a quantitative SR as we are much more flexible and are not necessarily trying to identify all existing literature on a topic. I suggest rephrasing.

Material and methods

The authors discuss studies from other LMIC settings outside of Africa, but it is unclear why these were searched for when the title is about Africa. I suggest that reference to these studies can be removed or needs to be clarified.

I would like more detail around how the authors decided if a study was rich or poor. They refer to levels of conceptual data and depth of analysis but not to thickness or richness of the data in the article (more traditional definitions of thick description). I do not feel that with the level of detail provided another author team could go out and make the same assessments for the included studies. Did they use a checklist? Only discussion? What was the cut-off point? Is it a yes or no or a sliding scale? Could they provide examples in a supplementary document of the decision process for a rich and poor study? I think this would be useful for future author teams to use the same approach.

Line 105- The authors write that they sampled poor studies…. But in the next section they write that they did not. Could 105 be changed to “we would have sampled…”

Reflexivity

See comment above on adding their standpoints on data saturation to the reflexivity section

Results

Line 140- Again I think reference to non-African studies can be removed.

Line 145- Can you please reorder the countries alphabetically or by N

Line 160- Were all the included studies single methods studies or did any use methods triangulation? Was this considered in your rich vs poor evaluation?

Line 209- Do you mean death sentence rather than death threat. A death threat seems odd here.

Line 253- Is there a word missing from the end of theme 5 (system?)

In general, throughout the results section I feel there is a need for qualifying language when it comes to participants. Often it is written as if all the participants were expressing the same point of view. I.e. line 266 Male HCWs are accustomed to…. This is a very broad statement and it would be difficult to see if all male HCWs felt this way. I feel it should be tempered by some, many, those male HCWs included in the sampled studies etc.

Discussion

In general, the authors argue that they have reached data saturation at the thematic level. However, they highlight that the new studies provided new insight and nuance to existing themes (extended, enriched). I would argue that this is a finding and that perhaps data saturation was not reached. I would like to seem some reflection on if they feel this new data was important enough that the review should be updated even though no new themes were added. Would the new extended and enriched data have implications for policy or practice especially around the new subgroups that were identified?

The authors provide useful information for QES groups contemplating updating a review.

I appreciate the reference to CERQual as I think this could be a useful measure. Did their findings move from low to moderate or high confidence with addition of the new studies? There has been one review to use CERQual on theoretical findings that will be published shortly (Cooper vaccination review with EPOC).

Conclusion

Line 427- Yes, no new themes were found but existing themes were enriched and extended in a very useful and interesting way and this is worth mentioning.

Other questions to consider but do not need to be added to the article:

Do the methods used in the included studies impact your certainty of reaching saturation? For example, if there are limited observation studies or a number of studies that only used a single method for data collection does this impact on a QES saturation?

Reviewer #2: I was extremely impressed by this article which is generally well-written and contributes well to the emerging evidence base on qualitative evidence synthesis updates. I had a few very specific points about phrasing and a few important points of accuracy and precision.

My main reservation is the quality of the Discussion – the authors touch on important issues but fail to engage sufficiently with the existing methodological literature. There is already a small body of evidence about sampling of qualitative evidence and also about conducting qualitative sensitivity analysis for quality as well as some recent work on richness and on relevance and on context. While I am not expecting the authors to engage equally with all of these methodological literatures the contribution of their article would be confirmed, extended and enhanced by pursuing at least one or two of these issues to take the contribution beyond an update case study.

Abstract:

rationale, triggers, and methods for updating is less clear cut. “is” should be “are”

“301 studies met the inclusion criteria. Rich findings from Africa were available in 82 studies; 146 studies were poor” This makes 228 studies - what happened to the remainder?

“careful appraisal of the new literature” – maybe “appraisal” is not a helpful term here because of its stronger technical association with quality assessment. Maybe “evaluation”?

Introduction:

“when new data just repeats was has already been found in previous data” – “what” not “was”

“As researchers are now adopting methods of quantitative research synthesis for QES and are discussing updating QES [5-7]” – This sentence is misleading – the point being made seems to be only relating to updating and I don’t see anything in the three cited references that demonstrate “methods of quantitative research synthesis” being used in QES. Something like: “As the increasing prevalence of QES is leading to an imperative to update QES that follows that previously encountered within quantitative research synthesis [5-7]” or similar.

In the Abstract you refer to “confirmed, extended, enriched, or refuted”. In the Introduction it is: “new, enrichment or modification” – it might be helpful to use a common terminology.

“We defined ‘rich’ studies “according to the depth of analysis reflected in the primary study authors interpretation of findings “” It might be helpful to acknowledge different approaches to operationalising “richness” of which the most developed is Ames, H., Glenton, C. & Lewin, S. Purposive sampling in a qualitative evidence synthesis: a worked example from a synthesis on parental perceptions of vaccination communication. BMC Med Res Methodol 19, 26 (2019). https://doi.org/10.1186/s12874-019-0665-4

“any additional subgroups not covered” – it is difficult at this point in the manuscript to understand what these “subgroups” might be – perhaps an example here e.g. are they population groups?

“We report the number of studies confirming our original findings, extending existing themes, and enriching our understanding of existing themes” What happened to “refuting”?

“We did not sample any poor papers.” State number of poor papers here “We did not sample any of the X poor papers”. (NB. We tend to use “sparse” instead of poor because it is unambiguously not about quality but I appreciate this may be personal preference).

“described a lot of positive effects” – “a lot of” is colloquial for a scientific paper.

“None of the studies refuted our theory, despite careful consideration within the team of possible adjustments being made.” I think that more could be made of this statement – i.e. what you were looking for and why you might not have found it.

I particularly liked your confirmed, extended enhanced descriptors as used when assessing the value of each revisited theme.

Methodologically there might have been a justification for a framework synthesis approach given the fact that you wanted to revisit the original analysis and see what an update might add. You could at least consider this methodological option within the article.

Discussion – this is disappointing as it identifies potential issues but does not relate to published literature– issues that could be considered include absence of refutational evidence (compare experience in meta-ethnography), differential definitions of richness (Ames paper above), the fact that poor quality studies [“these poor studies may fulfill other more immediate purposes such as informing specific programmes”] have been shown to add little conceptually to analyses but may add new contexts (Carroll & Booth). Also that new studies may automatically add new contexts temporally speaking (changing attitudes, contextual factors eg Covid) and that you may only know whether the new studies are worth including by doing this – i.e. you can’t predict whether an update will be worth doing in advance.

“In our previous review we had sampled thin studies” – the concept of “thin” studies is new to this paper – before you used “poor” Are they the same or different?

“It raises the question whether updating a QES which already has a substantive literature base should be done without very considerable thought if this is a sensible use of resources.” (“about whether” not “if”) This is a very simplistic and quantitative conclusion which is disappointingly at odds with the rest of the paper. For me, instead, this raises the question “what constitutes a “substantive literature base”?” It surely can’t just mean number of studies? So what else could it mean? And what might a review team do to decide whether theirs is “substantive” or not? Also might the type of analysis be a factor e.g. meta-ethnography versus thematic synthesis? I also wonder whether the Discussion would be enriched by some consideration of what saturation means in the context of primary research.

“good baseline understanding of the theory,” – Use of “baseline understanding of the theory” is a strange choice of expression – theories are qualitative and therefore don’t have a baseline. We would typically refer to “a good grounding in the theory”.

“then using the existing team is economical and sensible.” – economical, maybe but an alternative view about “sensible” is that for an interpretive work introducing new perspectives could enhance the analysis – looking at differences not similarities. Covered in Booth “disconfirming case” article.

“CERQual tool” – the approved designation is “GRADE-CERQual approach”. It is not described as a “tool” – it includes multiple processes. Also its purpose is not “formally assessing quality of themes” it is “for assessing how much confidence to place in findings from systematic reviews of qualitative research (or qualitative evidence syntheses)”.

Also the statement “designed for assessing individual descriptive findings rather than higher order analytical findings, and theory building” seems to imply that GRADE-CERQual cannot be used for the latter. A more accurate statement is that “and it is not currently known how well it performs when assessing higher order analytical findings, and theory building”. Also the reference should be to a paper in the newer GRADE-CERQual series not to the original PLOS article.

“Our updated QES of adherence to ART strengthened our theory” Again I think that you should utilise your previous descriptors of “confirmed, extended, and enriched” unless that is you are invoking the “crystalise” analogy of earlier which would imply a higher level of contribution.

6. PLOS authors have the option to publish the peer review history of their article (what does this mean?). If published, this will include your full peer review and any attached files.

Reviewer #1: **Yes: **Heather Melanie R Ames

Reviewer #2: **Yes: **Andrew Booth

---

## [Decision Letter · Decision Letter 1]

27 Sep 2021

Testing for saturation in qualitative evidence syntheses: An update of HIV adherence in Africa

PONE-D-21-10801R1

Dear Dr. Garner,

We’re pleased to inform you that your manuscript has been judged scientifically suitable for publication and will be formally accepted for publication once it meets all outstanding technical requirements.

Kind regards,

Sergi Fàbregues

Academic Editor

PLOS ONE

Reviewer's Responses to Questions

**Comments to the Author**

1. If the authors have adequately addressed your comments raised in a previous round of review and you feel that this manuscript is now acceptable for publication, you may indicate that here to bypass the “Comments to the Author” section, enter your conflict of interest statement in the “Confidential to Editor” section, and submit your "Accept" recommendation.

Reviewer #1: All comments have been addressed

2. Is the manuscript technically sound, and do the data support the conclusions?

Reviewer #1: (No Response)

3. Has the statistical analysis been performed appropriately and rigorously? 

Reviewer #1: (No Response)

4. Have the authors made all data underlying the findings in their manuscript fully available?

Reviewer #1: (No Response)

5. Is the manuscript presented in an intelligible fashion and written in standard English?

Reviewer #1: (No Response)

6. Review Comments to the Author

Reviewer #1: (No Response)

7. PLOS authors have the option to publish the peer review history of their article (what does this mean?). If published, this will include your full peer review and any attached files.

Reviewer #1: **Yes: **Heather Ames

---

## [Editor Report · Acceptance letter]

1 Oct 2021

PONE-D-21-10801R1 

Testing for saturation in qualitative evidence syntheses: 
An update of HIV adherence in Africa 

Dear Dr. Garner:

I'm pleased to inform you that your manuscript has been deemed suitable for publication in PLOS ONE. Congratulations! Your manuscript is now with our production department. 

Kind regards, 

on behalf of

Dr. Sergi Fàbregues 

Academic Editor

PLOS ONE